# Autonomous Air Traffic Controller: A Deep Multi-Agent Reinforcement Learning Approach

**Marc Brittain** [1]   **Peng Wei** [1]

## Abstract

Air traffic control is a real-time safety-critical decision making process in highly dynamic and stochastic environments. In today's aviation practice, a human air traffic controller monitors and directs many aircraft flying through its designated airspace sector. With the fast growing air traffic complexity in traditional (commercial airliners) and low-altitude (drones and eVTOL aircraft) airspace, an autonomous air traffic control system is needed to accommodate high density air traffic and ensure safe separation between aircraft. We propose a deep multi-agent reinforcement learning framework that is able to identify and resolve conflicts between aircraft in a high-density, stochastic, and dynamic en-route sector with multiple intersections and merging points. The proposed framework utilizes an actor-critic model, A2C that incorporates the loss function from Proximal Policy Optimization (PPO) to help stabilize the learning process. In addition we use a centralized learning, decentralized execution scheme where one neural network is learned and shared by all agents in the environment. We show that our framework is both scalable and efficient for large number of incoming aircraft to achieve extremely high traffic throughput with safety guarantee. We evaluate our model via extensive simulations in the BlueSky environment. Results show that our framework is able to resolve 99.97% and 100% of all conflicts both at intersections and merging points, respectively, in extreme high-density air traffic scenarios.

[1]Department of Aerospace Engineering, Iowa State University, Ames, IA, 50021, USA. Correspondence to: Marc Brittain <mwb@iastate.edu>, Peng Wei <pwei@iastate.edu>.

*Reinforcement Learning for Real Life (RL4RealLife) Workshop in the 36th International Conference on Machine Learning*, Long Beach, California, USA, 2019. Copyright 2019 by the author(s).

## 1. Introduction

### 1.1. Motivation

With the rapid increase in global air traffic and expected high density air traffic in specific airspace regions, to guarantee air transportation safety and efficiency becomes a critical challenge. Tactical air traffic control (ATC) decisions to ensure safe separation between aircraft are still being made by human air traffic controllers in en-route airspace sectors, which is the same as compared to 50 years ago (Council et al., 2014). Heinz Erzberger and his NASA colleagues first proposed autonomous air traffic control by introducing the Advanced Airspace Concept (AAC) to increase airspace capacity and operation safety by designing automation tools such as the Autoresolver and TSAFE to augment human controllers (Erzberger, 2005; Erzberger & Heere, 2010; Farley & Erzberger, 2007) in conflict resolution. Inspired by Erzberger, we believe that a fully automated ATC system is the ultimate solution to handle the high-density, complex, and dynamic air traffic in the future en-route and terminal airspace for commercial air traffic.

In recent proposals for low-altitude airspace operations such as UAS Traffic Management (UTM) (Kopardekar et al., 2016), U-space (Undertaking, 2017), and urban air mobility (Mueller, Kopardekar, and Goodrich, 2017), there is also a strong demand for an autonomous air traffic control system to provide advisories to these intelligent aircraft, facilitate on-board autonomy or human operator decisions, and cope with high-density air traffic while maintaining safety and efficiency (Air, 2015; Airbus, 2018; Google, 2015; Holden & Goel, 2016; Kopardekar, 2015; Mueller et al., 2017; Uber, 2018). According to the most recent study by Hunter and Wei (Hunter & Wei, 2019), the key to these low-altitude airspace operations is to design the autonomous ATC on structured airspace to achieve envisioned high throughput. Therefore, the critical challenge here is to design an autonomous air traffic control system to provide real-time advisories to aircraft to ensure safe separation both along air routes and at intersections of these air routes. Furthermore, we need this autonomous ATC system to be able to manage multiple intersections and handle uncertainty in real time.

To implement such a system, we need a model to perceive

the current air traffic situation and provide advisories to aircraft in an efficient and scalable manner. Reinforcement learning, a branch of machine learning, is a promising way to solve this problem. The goal in reinforcement learning is to allow an agent to learn an optimal policy by interacting with an environment. The agent is trained by first perceiving the state in the environment, selecting an action based on the perceived state, and receiving a reward based on this perceived state and action. By formulating the tasks of human air traffic controllers as a reinforcement learning problem, the trained agent can provide dynamic real-time air traffic advisories to aircraft with extremely high safety guarantee under uncertainty and little computation overhead.

Artificial intelligence (AI) algorithms are achieving performance beyond humans in many real-world applications today. An artificial intelligence agent called AlphaGo built by DeepMind defeated the world champion Ke Jie in three matches of Go in May 2017 (Silver & Hassabis, 2016). This notable advance in the AI field demonstrated the theoretical foundation and computational capability to potentially augment and facilitate human tasks with intelligent agents and AI technologies. To utilize such techniques, fast-time simulators are needed to allow the agent to efficiently learn in the environment. Until recently, there were no open-source high-quality air traffic control simulators that allowed for fast-time simulations to enable an AI agent to interact with. The air traffic control simulator, BlueSky, developed by TU Delft allows for realistic real-time air traffic scenarios and we decide to use this software as the environment and simulator for performance evaluation of our proposed framework (Hoekstra & Ellerbroek, 2016).

In this paper, a deep multi-agent reinforcement learning framework is proposed to enable autonomous air traffic separation in en-route airspace, where each aircraft is represented by an agent. Each agent will comprehend the current air traffic situation and perform online sequential decision making to select speed advisories in real-time to avoid conflicts at intersections, merging points, and along route. Our proposed framework provides another promising potential solution to enable an autonomous air traffic control system.

## 1.2. Related Work

Deep reinforcement learning has been widely explored in ground transportation in the form of traffic light control (Genders and Razavi, 2016; Liang, Du, Wang, and Han, 2018). In these approaches, the authors deal with a single intersection and use one agent per intersection to control the traffic lights. Our problem is similar to ground transportation in the sense we want to provide speed advisories to aircraft to avoid conflict, in the same way a traffic light advises cars to stop and go. The main difference with our problem is that we need to control the speed of each aircraft

to ensure there is no along route conflict. In our work, we represent each aircraft as an agent instead of the intersection to handle along route and intersection conflicts.

There have been many important contributions to the topic of autonomous air traffic control. One of the most promising and well-known lines of work is the Autoresolver designed and developed by Heinz Erzberger and his NASA colleagues (Erzberger, 2005; Erzberger & Heere, 2010; Farley & Erzberger, 2007). It employs an iterative approach, sequentially computing and evaluating candidate trajectories, until a trajectory is found that satisfies all of the resolution conditions. The candidate trajectory is then output by the algorithm as the conflict resolution trajectory. The Autoresolver is a physics-based approach that involves separate components of conflict detection and conflict resolution. It has been tested in various large-scale simulation scenarios with promising performance.

Strategies for increasing throughput of aircraft while minimizing delay in high-density sectors are currently being designed and implemented by NASA. These works include the Traffic Management Advisor (TMA) (Erzberger and Itoh, 2014) or Traffic Based Flow Management (TBFM), a central component of ATD-1 (Baxley, Johnson, Scardina, and Shay, 2016). In this approach, a centralized planner determines conflict free time-slots for aircraft to ensure separation requirements are maintained at the metering fix. Our algorithm also is able to achieve a conflict free metering fix by allowing the agents to learn a cooperative strategy that is queried quickly online during execution, unlike TBFM. Another main difference with our work is that our proposed framework is a decentralized framework that can handle uncertainty. In TMA or TBFM, once the arrival sequence is determined and aircraft are within the "freeze horizon" no deviation from the sequence is allowed, which could be problematic if one aircraft becomes uncooperative.

Multi-agent approaches have also been applied to conflict resolution (Wollkind, Valasek, and Ioerger, 2004). In this line of work, negotiation techniques are used to resolve identified conflicts in the sector. In our research, we do not impose any negotiation techniques, but leave it to the agents to derive negotiation techniques through learning and training.

Reinforcement learning and deep Q-networks have been demonstrated to play games such as Go, Atari and Warcraft, and most recently Starcraft II (Amato and Shani, 2010; Mnih, Kavukcuoglu, Silver, Graves, Antonoglou, Wierstra, and Riedmiller, 2013; Silver, Huang, Maddison, Guez, Sifre, Van Den Driessche, Schrittwieser, Antonoglou, Panneershelvam, Lanctot, et al., 2016; Vinyals, Ewalds, Bartunov, Georgiev, Vezhnevets, Yeo, Makhzani, Küttler, Agapiou, Schrittwieser, et al., 2017). The results from these papers show that a well-designed, sophisticated AI agent is capable

of learning complex strategies. It was also shown in previous work that a hierarchical deep reinforcement learning agent was able to avoid conflict and choose optimal route combinations for a pair of aircraft (Brittain and Wei, 2018).

Recently the field of multi-agent collision avoidance has seen much success in using a decentralized non-communicating framework in ground robots (Chen, Liu, Everett, and How, 2017; Everett, Chen, and How, 2018). In this work, the authors develop an extension to the policy-based learning algorithm (GA3C) that proves to be efficient in learning complex interactions between many agents. We find that the field of collision avoidance can be adapted to conflict resolution by considering larger separation requirements, so our framework is inspired by the ideas set forth by (Everett et al., 2018).

In this paper, the deep multi-agent reinforcement learning framework is developed to solve the separation problem for autonomous air traffic control in en-route dynamic airspace where we avoid the computationally expensive forward integration method by learning a policy that can be quickly queried. The results show that our framework has very promising performance.

The structure of this paper is as follows: in Section II, the background of reinforcement learning, policy based learning, and multi-agent reinforcement learning will be introduced. In Section III, the description of the problem and its mathematical formulation of deep multi-agent reinforcement learning are presented. Section IV presents our designed deep multi-agent reinforcement learning framework to solve this problem. The numerical experiments and results are shown in Section V, and Section VI concludes this paper.

## 2. Background

### 2.1. Reinforcement Learning

Reinforcement learning is one type of sequential decision making where the goal is to learn how to act optimally in a given environment with unknown dynamics. A reinforcement learning problem involves an environment, an agent, and different actions the agent can select in this environment. The agent is unique to the environment and we assume the agent is only interacting with one environment. If we let $t$ represent the current time, then the components that make up a reinforcement learning problem are as follows:

- $S$ - The state space $S$ is a set of all possible states in the environment

- $A$ - The action space $A$ is a set of all actions the agent can select in the environment

- $r(s_t, a_t)$ - The reward function determines how much

reward the agent is able to acquire for a given $(s_t, a_t)$ transition

- $\gamma \in [0,1]$ - A discount factor determines how far in the future to look for rewards. As $\gamma \to 0$, immediate rewards are emphasized, whereas, when $\gamma \to 1$, future rewards are prioritized.

$S$ contains all information about the environment and each element $s_t$ can be considered a snapshot of the environment at time $t$. The agent accepts $s_t$ and with this, the agent then selects an action, $a_t$. By selecting action $a_t$, the state is now updated to $s_{t+1}$ and there is an associated reward from making the transition from $(s_t, a_t) \to s_{t+1}$. How the state evolves from $s_t \to s_{t+1}$ given action $a_t$ is dependent upon the dynamics of the system, which is often unknown. The reward function is user defined, but needs to be carefully designed to reflect the goal of the environment.

From this framework, the agent is able to extract the optimal actions for each state in the environment by maximizing a cumulative reward function. We call the actions the agent selects for each state in the environment a policy. Let $\pi$ represent some policy and $T$ represent the total time for a given environment, then the optimal policy can be defined as follows:

$$\pi^* = \arg\max_{\pi} E[\sum_{t=0}^{T}(r(s_t, a_t)|\pi)]. \tag{1}$$

If we design the reward to reflect the goal in the environment, then by maximizing the total reward, we have obtained the optimal solution to the problem.

### 2.2. Policy-Based Learning

In this work, we consider a policy-based reinforcement learning algorithm to generate policies for each agent to execute. The advantage of policy-based learning is that these algorithms are able to learn stochastic policies, whereas value-based learning can not. This is especially beneficial in non-communicating multi-agent environments, where there is uncertainty in other agent's action. A3C (Mnih, Badia, Mirza, Graves, Lillicrap, Harley, Silver, and Kavukcuoglu, 2016), a recent policy-based algorithm, uses a single neural network to approximate both the policy (actor) and value (critic) functions with many threads of an agent running in parallel to allow for increased exploration of the state-space. The actor and critic are trained according to the two loss functions:

$$L_{\pi} = \log \pi(a_t, |s_t)(R_t - V(s_t)) + \beta \cdot H(\pi(s_t)) \tag{2}$$

$$L_v = (R_t - V(s_t))^2, \tag{3}$$

where in (2), the first term $\log \pi(a_t, |s_t)(R_t - V(s_t))$ reduces the probability of sampling an action that led to a

lower return than was expected by the critic and the second term, $\beta \cdot H(\pi(s_t))$ is used to encourage exploration by discouraging premature convergence to suboptimal deterministic polices. Here $H$ is the entropy and the hyperparameter $\beta$ controls the strength of the entropy regularization term. In (3), the critic is trained to approximate the future discounted rewards, $R_t = \sum_{i=0}^{k-1} \gamma^i r_{t+i} + \gamma^k V(s_{t+k})$.

One drawback of $L_\pi$ is that it can lead to large destructive policy updates and hinder the final performance of the model. A recent algorithm called Proximal Policy Optimization (PPO) solved this problem by introducing a new type of loss function that limits the change from the previous policy to the new policy (Schulman, Wolski, Dhariwal, Radford, and Klimov, 2017). If we let $r_t(\theta)$ denote the probability ratio and $\theta$ represent the neural network weights at time $t$, $r_t(\theta) = \frac{\pi_\theta(a_t|s_t)}{\pi_{\theta_{old}}(a_t|s_t)}$, the PPO loss function can be formulated as follows:

$$L^{\text{CLIP}}(\theta) =$$
$$E_t[\min(r_t(\theta)(A), clip(r_t(\theta), 1-\epsilon, 1+\epsilon)(A))], \quad (4)$$

where $A := R_t - V(s_t)$ and $\epsilon$ is a hyperparameter that determines the bound for $r_t(\theta)$. This loss function allows the previous policy to move in the direction of the new policy, but by limiting this change it is shown to lead to better performance (Schulman et al., 2017).

### 2.3. Multi-Agent Reinforcement Learning

In multi-agent reinforcement learning, instead of considering one agent's interaction with the environment, we are concerned with a set of agents that share the same environment (Bu, Babu, De Schutter, et al., 2008). Fig. 1 shows the progression of a multi-agent reinforcement learning problem. Each agent has its own goals that it is trying to achieve in the environment that is typically unknown to the other agents. In these types of problems, the difficulty of learning useful policies greatly increases since the agents are both interacting with the environment and each other. One strategy for solving multi-agent environments is Independent Q-learning (Tan, 1993), where other agents are considered to be part of the environment and there is no communication between agents. This approach often fails since each agent is operating in the environment and in return, creates learning instability. This learning instability is caused by the fact that each agent is changing its own policy and how the agent changes this policy will influence the policy of the other agents (Matignon, Laurent, and Le Fort-Piat, 2012). Without some type of communication, it is very difficult for the agents to converge to a good policy.

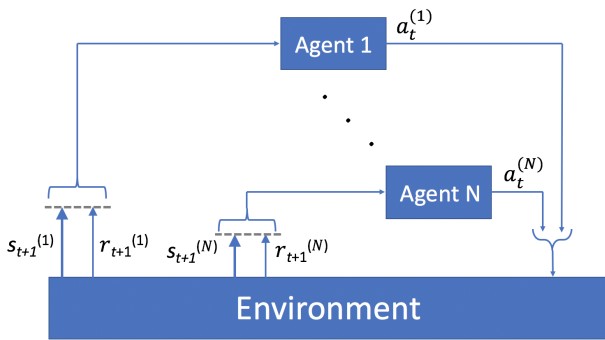

Figure 1. Progression of a multi-agent reinforcement learning problem.

## 3. Problem Formulation

In real world practice, air traffic controllers in en-route and terminal sectors are responsible for separating aircraft. In our research, we used the BlueSky air traffic control simulator as our deep reinforcement learning environment. We developed two challenging Case Studies: one with multiple intersections (Case Study 1) and one with a merging point (Case Study 2), both with high-density air traffic to evaluate the performance of our deep multi-agent reinforcement learning framework.

### 3.1. Objective

The objective in these Case Studies is to maintain safe separation between aircraft and resolve conflict for aircraft in the sector by providing speed advisories. In Case Study 1, three routes are constructed with two intersections so that the agents must navigate through the intersection with no conflicts. In Case Study 2, there are two routes that reach a merging point and continue on one route, so ensuring proper separation requirements at the merging point is a difficult problem to solve. In order to obtain the optimal solution in this environment, the agents have to maintain safe separation and resolve conflict and every time step in the environment. To increase the difficulty of the Case Studies and to provide a more realistic environment, aircraft enter the sector stochastically so that the agents need to develop a strategy instead of simply memorizing actions.

### 3.2. Simulation Settings

There are many settings we imposed to make these Case Studies feasible. For each simulation run, there is a fixed max number of aircraft. This is to allow comparable performance between simulation runs and to evaluate the final performance of the model. In BlueSky, the performance metrics of each aircraft type impose different constraints on the range of cruise speeds. We set all aircraft to be the same type, Boeing 747-400, in both Case Studies. We

also imposed a setting that all aircraft can not deviate from their route. The final setting in the Case Studies is the desired speed of each aircraft. Each aircraft has the ability to select three different desired speeds: minimum allowed cruise speed, current cruise speed, and maximum allowed cruise speed which is defined in the BlueSky simulation environment.

### 3.3. Multi-Agent Reinforcement Learning Formulation

Here we formulate our BlueSky Case Study as a deep multi-agent reinforcement learning problem by representing each aircraft as an agent and define the state space, action space, termination criteria and reward function for the agents.

#### 3.3.1. STATE SPACE

A state contains all the information the AI agent needs to make decisions. Since this is a multi-agent environment, we needed to incorporate communication between the agents. To allow the agents to communicate, the state for a given agent also contains state information from the $N$-closest agents. We follow a similar state space definition as in (Everett et al., 2018), but instead we use half of the loss of separation distance as the radius of the aircraft. In this way the sum of the radii between two aircraft is equal to the loss of separation distance. The state information includes distance to the goal, aircraft speed, aircraft acceleration, distance to the intersection, a route identifier, and half the loss of separation distance for the $N$-closest agents, where the position for a given aircraft can be represented as (distance to the goal, route identifier). We also included the distance to the $N$-closest agents in the state space of the agents and the full loss of separation distance. From this, we can see that the state space for the agents is constant in size, since it only depends on the $N$-closest agents and does not scale as the number of agents in the environment increase. Fig. 2 shows an example of a state in the BlueSky environment.

We found that defining which $N$-closest agents to consider is very important to obtain a good result since we do not want to add irrelevant information in the state space. For example, consider Fig. 2. If the ownship is on $R_1$ and one of the closest aircraft on $R_3$ has already passed the intersection, there is no reason to include its information in the state space of the ownship. We defined the following rules for the aircraft that are allowed to be in the state of the ownship:

- aircraft on conflicting route must have not reached the intersection

- aircraft must either be on the same route or on a conflicting route.

By utilizing these rules, we eliminated useless information

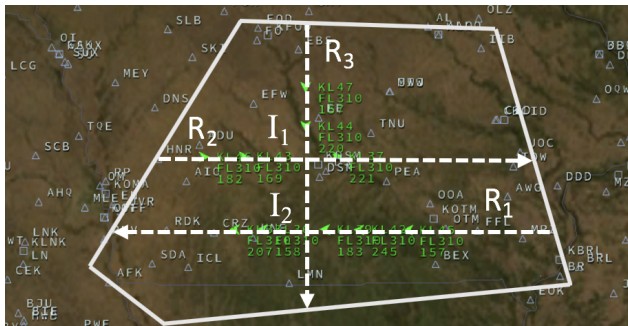

*Figure 2.* BlueSky sector designed for our Case Study. Shown is Case Study 1 for illustration: there are three routes, $R_1$, $R_2$, and $R_3$, along with two intersections, $I_1$ and $I_2$.

which we found to be critical in obtaining convergence to this problem.

If we consider Fig. 2 as an example, we can acquire all of the state information we need from the aircraft. If we let $I^{(i)}$ represent the distance to the goal, aircraft speed, aircraft acceleration, distance to the intersection, route identifier, and half the loss of separation distance of aircraft $i$, the state will be represented as follows:

$$s_t^o = (I^{(o)}, d^{(1)}, \text{LOS}(o, 1), d^{(2)}, \text{LOS}(o, 2)..., d^{(n)},$$
$$\text{LOS}(o, n), I^{(1)}, I^{(2)}, ..., I^{(n)}),$$

where $s_t^o$ represents the ownship state, $d^{(i)}$ represents the distance from ownship to aircraft $i$, $\text{LOS}(o, i)$ represents the loss of separation distance between aircraft $o$ and aircraft $i$, and $n$ represents the number of closest aircraft to include in the state of each agent. By defining the loss of separation distance between two aircraft in the state space, the agents should be able to develop a strategy for non-uniform loss of separation requirements for different aircraft types. In this work we consider the standard uniform loss of separation requirements and look to explore this idea in future work.

#### 3.3.2. ACTION SPACE

All agents decide to change or maintain their desired speed every 12 seconds in simulation time. The action space for the agents can be defined as follows:

$$A_t = [v_{\min}, v_{t-1}, v_{\max}],$$

where $v_{min}$ is the minimum allowed cruise speed (decelerate), $v_{t-1}$ is the current speed of the aircraft (hold), and $v_{max}$ is the maximum allowed cruise speed (accelerate).

#### 3.3.3. TERMINAL STATE

Termination in the episode was achieved when all aircraft had exited the sector:

$$N_{\text{aircraft}} = 0.$$

### 3.3.4. REWARD FUNCTION

The reward function for the agents were all identical, but locally applied to encourage cooperation between the agents. If two agents were in conflict, they would both receive a penalty, but the remaining agents that were not in conflict would not receive a penalty. Here a conflict is defined as the distance between any two aircraft is less than 3 nmi. The reward needed to be designed to reflect the goal of this paper: safe separation and conflict resolution. We were able to capture our goals in the following reward function for the agents:

$$
r_t = \begin{cases} -1 & \text{if } d_o^c < 3 \\ -\alpha + \beta \cdot d_o^c & \text{if } d_o^c < 10 \text{ and } d_o^c \geq 3 \\ 0 & \text{otherwise} \end{cases},
$$

where $d_o^c$ is the distance from the ownship to the closest aircraft in nautical miles, and $\alpha$ and $\beta$ are small, positive constants to penalize agents as they approach the loss of separation distance. By defining the reward to reflect the distance to the closest aircraft, this allows the agent to learn to select actions to maintain safe separation requirements.

## 4. Solution Approach

To solve the BlueSky Case Studies, we designed and developed a novel deep multi-agent reinforcement learning framework called the Deep Distributed Multi-Agent Reinforcement Learning framework (DD-MARL). In this section, we introduce and describe the framework, then we explain why this framework is needed to solve this Case Study.

To formulate this environment as a deep multi-agent reinforcement learning problem, we utilized a centralized learning with decentralized execution framework with one neural network where the actor and critic share layers of same the neural network, further reducing the number of trainable parameters. By using one neural network, we can train a model that improves the joint expected return of all agents in the sector, which encourages cooperation between the agents. We utilized the synchronous version of A3C, A2C (advantage actor critic) which is shown to achieve the same or better performance as compared to the asynchronous version (Schulman et al., 2017). We also adapted the A2C algorithm to incorporate the PPO loss function defined in (6), which we found led to a more stable policy and resulted in better final performance. We follow a similar approach to (Everett et al., 2018) to split the state into the two parts: ownship state information and all other information, which we call the local state information, $s^{local}$. We then encode the local state information using a fully connected layer before combining the encoded state with the ownship state information. From there, the combined state is sent through

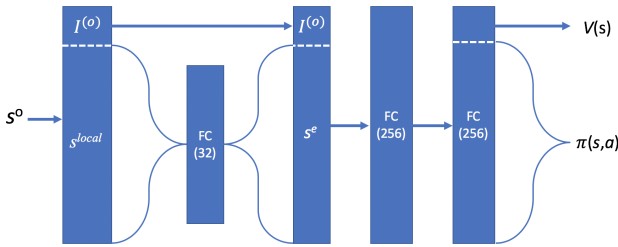

*Figure 3.* Illustration of the neural network architecture for A2C with shared layers between the actor and critic. Each hidden layer is a fully connected (FC) layer with 32 nodes for the encoded state and 256 nodes for the last two layers.

two fully connected layers and produces two outputs: the policy and value for a given state. Fig. 3 shows an illustration of the the neural network architecture. With this framework, we can implement the neural network to all aircraft, instead of having a specified neural network for all individual aircraft. In this way, the neural network is acting as a centralized learner and distributing knowledge to each aircraft. The neural network's policy is distributed at the beginning of each episode and updated at the end of each episode which reduces the amount of information that is sent to each aircraft, since sending an updated model during the route could be computationally expensive. In this formulation, each agent has identical neural networks, but since they are evolving different states their actions can be different.

It is also important to note that this framework is invariant to the number of aircraft. When observing an en-route sector, aircraft are entering and exiting which creates a dynamic environment with varying number of aircraft. Since our approach does not depend on the number of aircraft, our framework can handle any number of aircraft arriving based on stochastic inter-arrival times.

## 5. Numerical Experiments

### 5.1. Interface

To test the performance of our proposed framework, we utilized the BlueSky air traffic control simulator. This simulator is built around python so we were able to quickly obtain the state space information of all aircraft[1]. By design, when restarting the simulation, all objectives were the same: maintain safe separation and sequencing, resolve conflicts, and minimize delay. Aircraft initial positions and available speed changes did not change between simulation runs.

---

[1]Code will be made available at https://github.com/marcbrittain

## 5.2. Environment Setting

For each simulation run in BlueSky, we discretized the environment into episodes, where each run through the simulation counted as one episode. We also introduced a time-step, $\Delta t$, so that after the agents selected an action, the environment would evolve for $\Delta t$ seconds until a new action was selected. We set $\Delta t$ to 12 seconds to allow for a noticeable change in state from $s_t \rightarrow s_{t+1}$ and to check the safe-separation requirements at regular intervals.

There were many different parameters that needed to be tuned and selected for the Case Studies. We implemented the adapted A2C concept mentioned earlier, with two hidden layers consisting of 256 nodes. The encoding layer for the $N$-closest aircraft state information consisted of 32 nodes and we used the ReLU activation function for all hidden layers. The output of the actor used a Softmax activation function and the output of the critic used a Linear activation function. Other key parameter values included: learning rate $lr = 0.0001$, $\gamma = 0.99$, $\epsilon = 0.2$, $\alpha = 0.1$, $\beta = 0.005$, and we used the Adam optimizer for both the actor and critic loss (Kingma and Ba, 2014).

### 5.3. Case Study 1: Three routes with two intersections

In this Case Study, we considered three routes with two intersections as shown in Fig. 2. In our DD-MARL framework, the single neural network is implemented on each aircraft as they enter the sector. Each agent is then able to select its own desired speed which greatly increases the complexity of this problem since the agents need to learn how to cooperate in order to maintain safe-separation requirements. What also makes this problem interesting is that each agent does not have a complete representation of the state space since only the ownship (any given agent) state information and the $N$-closest agent state information are included.

### 5.4. Case Study 2: Two routes with one merging point

This Case Study consisted of two routes merging to one single point and then following one route thereafter (see Fig. 4. This poses another critical challenge for an autonomous air traffic control system that is not present in Case Study 1: merging. When merging to a single point, safe separation requirements need to be maintain both before the merging point and after. This is particularly challenging since there is high density traffic on each route that is now combining to one route. Agents need to carefully coordinate in order to maintain safe separation requirements after the merge point.

In both Case Studies there were 30 total aircraft that entered the airspace following a uniform distribution over 4, 5, and 6 minutes. This is an extremely difficult problem to solve because the agents cannot simply memorize actions, the

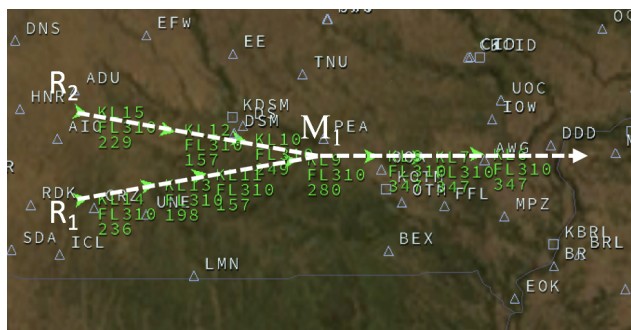

*Figure 4.* Case Study 2: two routes, $R_1$ and $R_2$ merge to a single route at $M_1$.

*Table 1.* Performance of the policy tested for 200 episodes.

| CASE STUDY | MEAN | MEDIAN |
|---|---|---|
| 1 | $29.99 \pm 0.141$ | 30 |
| 2 | 30 | 30 |

agents need to develop a strategy in order to solve the problem. We also included the 3 closest agents state information in the state of the ownship. All other agents are not included in the state of the ownship. The episode terminated when all 30 aircraft had exited the sector, so the optimal solution in this problem is 30 goals achieved. Here we define goal achieved as an aircraft exiting it the sector without conflict.

### 5.5. Algorithm Performance

In this section, we analyze the performance of DD-MARL on the Case Studies. We allowed the AI agents to train for 20,000 episodes and 5,000 episodes for Case Study 1 and Case Study 2, resepctively. We then evaluated the final policy for 200 episodes to calculate the mean and standard deviation along with the median to evaluate the performance of the final policy as shown in Table 1[2]. We can see from Fig. 5 that for Case Study 1, the policy began converging to a good policy by around episode 7,500, then began to further refine to a near optimal policy for the remaining episodes. For Case Study 2, we can see from Fig. 6 that a near optimal policy was obtained in only 2,000 episodes and continued to improve through the remainder of the 3,000 episodes. Training for only 20,000 episodes (as required in Case Study 1) is computationally inexpensive as it equates to less than 4 days of training. We suspect that this is due to the approach of distributing one neural network to all aircraft and by allowing shared layers between the actor and critic.

---

[2] A video of the final converged policy can be found at https://www.youtube.com/watch?v=sjRGjiRZWxg and https://youtu.be/NvLxTJNd-q0

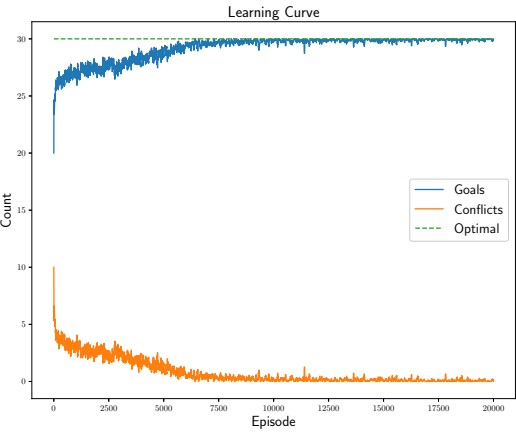

*Figure 5.* Learning curve of the DD-MARL framework for Case Study 1. Results are smoothed with a 30 episode rolling average for clarity.

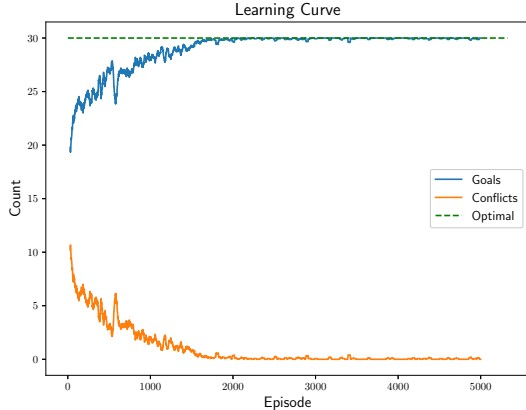

*Figure 6.* Learning curve of the DD-MARL framework for Case Study 2. Results are smoothed with a 30 episode rolling average for clarity.

We can see from Table 1, that on average we obtained a score of 29.99 throughout the 200 episode testing phase for Case Study 1 and 30 for Case Study 2. This equates to resolving conflict 99.97% at the intersections, and 100% at the merging point. Given that this is a stochastic environment, we speculate that there could be cases where there is an orientation of aircraft where the 3 nmi loss of separation distance can not be achieved, and in such cases we would alert human ATC to resolve this type of conflict. The median score removes any outliers from our testing phase and we can see the median score is optimal for both Case Study 1 and Case Study 2.

## 6. Conclusion

We built an autonomous air traffic controller to ensure safe separation between aircraft in high-density en-route airspace sector. The problem is formulated as a deep multi-agent reinforcement learning problem with the actions of selecting desired aircraft speed. The problem is then solved by using the DD-MARL framework, which is shown to be capable of solving complex sequential decision making problems under uncertainty. According to our knowledge, the major contribution of this research is that we are the first research group to investigate the feasibility and performance of autonomous air traffic control with a deep multi-agent reinforcement learning framework to enable an automated, safe and efficient en-route airspace. The promising results from our numerical experiments encourage us to conduct future work on more complex sectors. We will also investigate the feasibility of the autonomous air traffic controller to replace human air traffic controllers in ultra dense, dynamic and complex airspace in the future.

## Acknowledgements

We would like to thank Xuxi Yang, Guodong Zhu, Josh Bertram, Priyank Pradeep, and Xufang Zheng, whose input and discussion helped in the success of this work.

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
