# OpenReview forum: "Autonomous Air Traffic Controller: A Deep Multi-Agent Reinforcement Learning Approach"
_ICML.cc/2019/Workshop/RL4RealLife — RL4RealLife 2019_

### Official Review · AnonReviewer1 · 2019-05-23
**This paper shows several case studies to use deep multi-agent RL to control the autonomous air traffic system utilizing A2C and PPO.**

**Rating:** 3
**Confidence:** 3

**Review:**

Pros:
- The problem of proposed work is interesting and has not been tackled by RL yet. Although the author used existing method, the application is novel and refreshing.
- The paper is well written and easy to follow.

Cons:
- Background and related work explaining basic of RL methods is redundant.
- It is not clear that the reason why the author choose A3C, A2C among many alternatives.
- The experimental results are not enough and cannot justify the efficiency of proposed method without other baselines.

---

### Official Review · AnonReviewer2 · 2019-05-25
**I think the paper meet the demand of acceptance due to the comprehensive consideration of complex conditions in real life for airline conflicting.**

**Rating:** 5
**Confidence:** 5

**Review:**

In this paper, authors proposed a deep multi-agent reinforcement learning framework which adapts to complex condition of aircraft on route conflicting. Due to the interaction effect of different agents, authors consider the nearest N agents according to the distance and full loss of separation distance. The reflection of this is dynamic reward function. For the action, authors design a balanced speed with range of minimum and maximum allowed cruise speed.
From the overall point of view, the framework could accommodate the complicated and changeable environment which integrates dynamic change among different agents and the relation among current agent and other agents. In the real world, there would be multiple agents with disparate conditions which would show effects on the decision of flying at each time point. Authors also take that in consideration.
However, there are also some details need to be specified or improved.
Firstly, authors emphasize the aircrafts on conflicting route should not be reached the interaction due to the meaningless of considering already passed intersection. However, in real world, aircrafts would be interacted multiple times during the whole airline. It depends on the degree of deviation and their subsequent changes which interact with environment. So this hypothesis would be somehow a little strong.
Secondly, as a possible aspect for improvement, some external environmental change caused by some other settings such as clouds and airflow would also influence the actions and environment in real air traffic scene. So it would be better to incorporate factors which reflect these scenes.
Finally, for the settings, aircraft should have the ability to select speed in the range between minimum allowed cruise speed and maximum allowed cruise speed instead of just three desired speeds. In real life, speed would be changeable according to the environment. So there should be a rule for describing the change such as weight decaying or some other methods.

---

### Decision · Program_Chairs · 2019-05-28

Accept